# Inhibition of the FGF/FGFR System Induces Apoptosis in Lung Cancer Cells via c-Myc Downregulation and Oxidative Stress

**DOI:** 10.3390/ijms21249376

**Published:** 2020-12-09

**Authors:** Arianna Giacomini, Sara Taranto, Sara Rezzola, Sara Matarazzo, Elisabetta Grillo, Mattia Bugatti, Alessia Scotuzzi, Jessica Guerra, Martina Di Trani, Marco Presta, Roberto Ronca

**Affiliations:** 1Department of Molecular and Translational Medicine, University of Brescia, 11, 25123 Brescia, Italy; sara.taranto@unibs.it (S.T.); sara.rezzola@unibs.it (S.R.); sara.matarazzo8@gmail.com (S.M.); elisabetta.grillo@unibs.it (E.G.); mattia.bugatti@unibs.it (M.B.); scotuzzina@gmail.com (A.S.); Jessica.guerra@unibs.it (J.G.); Martina.Di_Trani@humanitasresearch.it (M.D.T.); marco.presta@unibs.it (M.P.); 2ASST Spedali Civili di Brescia, 11, 25123 Brescia, Italy; 3Humanitas Cancer Center, Humanitas Clinical and Research Center, 56, 20089 Milan, Italy; 4Italian Consortium for Biotechnology (CIB), Unit of Brescia, 25123 Brescia, Italy

**Keywords:** FGFR1, FGF, fibroblast growth factor, lung cancer, FGF trap, tyrosine kinase inhibitor, squamous cell carcinoma

## Abstract

Lung cancer represents an extremely diffused neoplastic disorder with different histological/molecular features. Among the different lung tumors, non-small-cell lung cancer (NSCLC) is the most represented histotype, characterized by various molecular markers, including the expression/overexpression of the fibroblast growth factor receptor-1 (FGFR1). Thus, FGF/FGFR blockade by tyrosine kinase inhibitors (TKi) or FGF-ligand inhibitors may represent a promising therapeutic approach in lung cancers. In this study we demonstrate the potential therapeutic benefit of targeting the FGF/FGFR system in FGF-dependent lung tumor cells using FGF trapping (NSC12) or TKi (erdafitinib) approaches. The results show that inhibition of FGF/FGFR by NSC12 or erdafitinib induces apoptosis in FGF-dependent human squamous cell carcinoma NCI-H1581 and NCI-H520 cells. Induction of oxidative stress is the main mechanism responsible for the therapeutic/pro-apoptotic effect exerted by both NSC12 and erdafitinib, with apoptosis being abolished by antioxidant treatments. Finally, reduction of c-Myc protein levels appears to strictly determine the onset of oxidative stress and the therapeutic response to FGF/FGFR inhibition, indicating c-Myc as a key downstream effector of FGF/FGFR signaling in FGF-dependent lung cancers.

## 1. Introduction

Lung tumors are the leading cause of cancer-related mortality worldwide, making up almost 25% of all cancer deaths and giving rise to more than 1 million deaths annually (www.cancer.org). Non-small-cell lung cancer (NSCLC) accounts for ~80% of cases and is divided roughly equally into two main histological types, adenocarcinomas and squamous cell carcinomas (SCC), the latter being strongly associated with smoking. Traditional chemotherapeutic approaches make only a small difference in outcome for NSCLC patients, so there has been considerable interest in and attempts to develop new and targeted therapies based on lung tumor biology [1].

Fibroblast growth factors (FGFs) represent a family of 22 structurally related growth factors fostering a wide range of biological effects, including mitogenesis and cellular differentiation, as well as increased cell motility and invasiveness in specific contexts [2]. FGFs mediate their biological responses by binding to cell surface high affinity tyrosine kinase (TK) FGF receptors (FGFRs), designated FGFR-1 to -4 [3]. An aberrant ligand-dependent or ligand-independent activation of the FGF/FGFR system due to FGFR activating mutations/overexpression, chromosomal rearrangements, or FGF overexpression has been found in different human tumors [4]. On this basis, the blockade of the FGF/FGFR system has been widely explored as a promising therapeutic approach in different types of cancer [5]. Tyrosine kinase inhibitors (TKi) with a broad/non-selective profile have been used with considerable side and off-target effects and nowadays more FGFR-selective inhibitors are available and have entered the clinical evaluation [5,6,7]. Furthermore, FGF-trap approaches have been developed as FGFR-mimics [8] or small molecules [9,10] to prevent the activation of the receptor by the ligand.

In lung cancer, FGFR1 has been found amplified in about 6% of patients classified either as SCLC or NSCLC, and FGFR1 amplifications are predominant (~17%) in patients with SCC followed by SCLC (~6%) and adenocarcinoma (~1%) [11,12,13]. The analysis of high-resolution genomic profiles of 77 lung adenocarcinomas and 155 lung SCC have shown that amplifications of the chromosomal region 8p12 were specific to SCC and that this amplification was centered on FGFR1 [14]. FGFR1 amplification appears to confer lung tumor dependence on FGF/FGFR signaling via an FGF-mediated autocrine mechanism of activation [15,16]. This promotes cancer cell survival, chemotherapy resistance and growth of lung tumor cell lines [14,17]. Accordingly, lung cancer cell lines (like NCI-H1581 and NCI-H520 cells) have been identified that harbor amplification of FGFR1 and are highly sensitive to FGFR inhibitors in vitro and in vivo [18,19].

In this study we describe the effect and the molecular mechanism of action of a prototypical FGFR-selective TKi (erdafitinib, JNJ-42756493) [20] and of the pan FGF-trap small molecule NSC12 (that specifically binds and sequesters all members of the human and murine FGF family) [10,21] in two different FGFR1-amplified lung SCC cell lines. In keeping with our previous findings in multiple myeloma models [22], data herein reported reveal that treatment with both erdafitinib and NSC12 induces oxidative-stress-mediated apoptosis in FGF-dependent lung cancer cells. These effects correlate to reduced c-Myc levels, indicating that c-Myc modulation may determine the therapeutic response to FGF/FGFR inhibitors in lung cancer.

## 2. Results

### 2.1. FGF/FGFR Inhibition Triggers Apoptosis in FGF/FGFR-Dependent Lung Cancer Cells

Human FGF-dependent H1581 cells are characterized by FGFR1 amplification [23] and expression of various members of the FGF family (Appendix A) that lead to FGFR activation via an autocrine mechanism of action (see below). On this basis, in order to assess the effects of FGF/FGFR inhibition in lung cancer cells, H1581 cells were exposed for 48 h to increasing doses of the pan FGF trap NSC12 or the FGFR specific TKi erdafitinib. Treatment with NSC12 or erdafitinib significantly reduced tumor cell proliferation, with an IC50 equal to 2.6 μM and 14 nM, respectively (Figure 1A), and induced apoptosis (Figure 1B). Accordingly, both compounds were able to cause the inhibition of FGFR activation, as shown by Western blot analysis of the cell extracts probed with a pan anti-phospho-FGFR antibody (Figure 1C).

Downregulation of the levels of c-Myc protein has been described as a key mechanism in the therapeutic response to FGF/FGFR inhibitors [24]. We have recently demonstrated that the reduction of c-Myc levels induced by FGF/FGFR inhibition triggers DNA damage and oxidative-stress-mediated apoptosis in multiple myeloma cells [10,22]. Accordingly, inhibition of FGFR activation by NSC12 or erdafitinib was paralleled by a significant decrease of the levels of c-Myc and its target genes *PRDX3*, *PRDX4*, *MCMC2* and *EIF3B* (Appendix A) and caused DNA damage in H1581 cells, as revealed by an increase of H2AX (γ-H2AX) protein phosphorylation and of cleaved PARP (Figure 1C). Together, these data suggest that inhibition of FGFR activation and down-modulation of c-Myc protein may induce apoptosis in lung cancer cells as a consequence of oxidative-stress-induced DNA damage.

### 2.2. Apoptosis Upon FGF/FGFR Inhibition is Induced by Oxidative Stress

In order to investigate the onset of oxidative stress in lung cancer cells upon FGF/FGFR inhibition, we assessed the production of reactive oxygen species (ROS) in H1581 cells after treatment with NSC12 or erdafitinib. As shown in Figure 2A and Appendix A, both inhibitors induced cytoplasmic ROS production paralleled by mitochondrial depolarization in H1581 cells, as demonstrated by the significant increase of DCFDA-positive and TMRE-negative cells, respectively. Of note, at variance with multiple myeloma cells [22], FGF/FGFR inhibition did not induce mitochondrial ROS production in H1581 cells, as assessed with the specific mitochondrial ROS probe Mitosox (Figure 2A). Treatment with the antioxidant vitamin E rescued H1581 cells from both NSC12 and erdafitinib-induced mitochondrial depolarization and apoptosis, indicating that oxidative stress is directly responsible for lung cancer cell death (Figure 2A and Appendix A). In keeping with the production of cytoplasmic ROS and the lack of mitochondrial ROS, the overexpression of cytoplasmic catalase, but not of mitochondrial catalase, significantly reduced H1581 cell death after treatment with both FGF/FGFR inhibitors (Figure 2B). Based on these data showing a shared mechanism of action for both FGF trapping and FGFR TKi approaches, the FGF trap molecule NSC12 was used for the next experiments.

### 2.3. Fgf Trapping-Mediated C-Myc Modulation and Consequent Oxidative Stress Are Specific for Fgf-Dependent Lung Cancer Cells

In order to investigate whether the induction of oxidative stress by FGF/FGFR inhibition is a mechanism specific for FGF-dependent lung cancers, we tested the effect of NSC12 on two other human lung cancer cell lines: FGF-dependent H520 cells and FGF-independent HCC827 cells. H520 cells, like H1581 cells, are characterized by FGFR1 amplification and autocrine FGF stimulation (Appendix A) [19], whereas HCC827 cells are adenocarcinoma cells that harbor a tumor driving mutation in the TK domain of EGFR which makes these cells independent from the FGF/FGFR system, notwithstanding their FGF/FGFR expression (Appendix A) [25]. As previously reported [21], NSC12 significantly reduced the proliferation of FGF-dependent H520 cells, but not of FGF-independent HCC827 cells (Figure 3A). Interestingly, NSC12 inhibited FGFR activation in both cell lines but resulted in a significant decrease of c-Myc levels and its target genes only in H520 cells (Figure 3B and Appendix A) that was paralleled by mitochondrial and cytoplasmic ROS production and apoptosis (Figure 3C). In keeping with the lack of c-Myc modulation in NSC12-treated HCC827 cells, neither ROS production nor apoptosis were observed in these cells (Figure 3C). Again, as observed in H1581 cells, inhibition of ROS production by the antioxidant vitamin E rescued H520 cells from NSC12-induced mitochondrial depolarization and apoptosis, thus confirming that oxidative stress is directly responsible for lung cancer cell death upon FGF/FGFR inhibition (Figure 4A and Appendix A). Notably, despite the presence of mitochondrial ROS in NSC12-treated H520 cells (Figure 4A), the overexpression of cytoplasmic, but not mitochondrial catalase significantly reduced H520 cell death after treatment with NSC12 (Figure 4B). These data suggest that the induction of cytoplasmic oxidative stress represents the main mechanism responsible for lung cancer cell death following FGF/FGFR inhibition, with the alteration of mitochondrial respiration a mere consequence of cytoplasmic ROS production.

### 2.4. FGF Trapping Inhibits Tumor Growth and Induces Oxidative Stress and DNA Damage in Vivo in FGF-Dependent Lung Cancer

In order to investigate the presence of a c-Myc/oxidative stress/DNA damage axis in vivo after FGF inhibition, we took advantage of tumor samples from FGF-dependent (H520) and FGF-independent (TC-1) xenograft models already established in a previous work [21]. Similar to FGF-independent HCC827 cells, no significant increase of oxidative stress and apoptosis were observed in TC-1 cells treated in vitro with NSC12 when compared to FGF-dependent H520 cells (Appendix A).

In vivo results demonstrated that parenteral and oral treatment with the FGF trap molecule NSC12 inhibited tumor angiogenesis, induced tumor cell apoptosis and resulted in impaired growth of H520, but not TC-1 tumor xenografts [21] (Figure 5). In keeping with the in vitro data, immunohistochemical analysis of H520 tumors from NSC12-treated mice revealed a significant down-modulation of the levels of c-Myc protein and the presence of both oxidative stress and DNA damage, as assessed by nitrotyrosine and γ-H2AX immunostainings, respectively (Figure 5A). At variance, neither significant modulation of c-Myc levels nor oxidative stress and DNA damage were observed in TC-1 xenografts from NSC12-treated mice (Figure 5B). These data confirm that c-Myc reduction, paralleled by oxidative stress and DNA damage induction, represents a key mechanism for the antitumor activity of FGF/FGFR inhibition in FGF-dependent lung cancers.

## 3. Discussion

Lung cancer is a life-threatening disease with high incidence worldwide and is often associated with bad prognosis. The majority of lung cancers are represented by NSCLC which are extremely heterogeneous and characterized by different biomarkers and different therapeutic approaches [26,27,28]. Among them, SCCs are often characterized by FGFR1 amplifications that make these lung tumors dependent on FGF ligands in vitro and in vivo [29,30], suggesting FGF blockade by FGF trapping or FGFR inhibition by TK inhibitors might represent promising therapeutic strategies for lung cancer treatment.

Here, we investigated the effects of FGF/FGFR inhibition in lung SCC by using both FGF trapping and FGFR TKi approaches. Our data demonstrate that FGF/FGFR inhibition induces apoptosis in lung cancer cells, supporting the notion that the FGF/FGFR system plays an important role in the survival of these cells. Further, data herein reported provide novel insights into the mechanisms by which FGF/FGFR inhibition triggers lung cancer cell apoptosis.

We have recently demonstrated that FGF/FGFR blockade induces c-Myc degradation which is essential to trigger oxidative-stress-mediated apoptosis in multiple myeloma cells [22]. In the present work, we extend these observations by demonstrating that the apoptotic process triggered by FGF/FGFR inhibition in lung cancer cells is induced by oxidative stress. Indeed, both vitamin E treatment and cytosolic catalase overexpression were able to rescue lung cancer cells from apoptosis. Moreover, our data indicate that the modulation of c-Myc protein levels strictly correlates with oxidative stress and apoptosis in lung cancer. Indeed, reduced levels of c-Myc protein upon FGF/FGFR inhibition were paralleled by ROS production, mitochondrial depolarization and DNA damage both in vitro and in vivo. In order to better investigate these mechanisms in lung cancer, we took advantage from FGF-dependent and FGF-independent lung tumor models already investigated in our laboratory [21]. Interestingly, at variance with FGF-dependent lesions, neither oxidative stress nor DNA damage or apoptosis were observed in FGF-independent xenografts where c-Myc levels were not modulated by FGF trapping. These data indicate that c-Myc protein may act as a key downstream effector for FGF/FGFR signaling in FGF-dependent cancers. Thus, the lack of sensitivity to FGFR inhibition observed in FGF-independent cancers may result from its incapacity to lead to c-Myc protein downregulation, which is prevented by alternative oncogenic molecular pathways. Further experiments are required to support this hypothesis.

Altogether, our findings indicate that FGF/FGFR blockade, c-Myc down-modulation, oxidative stress and apoptosis are strictly interconnected molecular processes in FGF-dependent lung tumors. Several pieces of evidence suggest the intracellular redox status may represent a promising therapeutic target in cancer. Indeed, oxidative-stress-inducing agents have been approved for the treatment of solid and hematological malignancies [31]. Accordingly, our data indicate that the induction of ROS production by FGF/FGFR inhibitors may represent a novel approach in FGF-dependent lung cancer.

As demonstrated by us and others [22,24], our data confirm the pivotal role of c-Myc modulation in the therapeutic response to FGF/FGFR inhibitors. Interestingly, MYC is expressed in 40% of FGFR1-amplified tumors and tumor cells co-expressing MYC and FGFR1 are more sensitive to FGFR inhibition [32]. These data suggest that patients with FGFR1-amplification and MYC-overexpression may benefit from anti-FGF/FGFR therapies and that the evaluation of c-Myc levels may serve as a biomarker for the response to FGF/FGFR inhibition in FGF-addicted cancers.

Finally, our findings provide a warning about antioxidant intake during antitumor therapies [23,33]. Indeed, vitamin E (and catalase overexpression) significantly reduced oxidative stress and apoptosis upon FGF/FGFR inhibition, thus impairing the antitumor effect of both NSC12 and erdafitinib. Additional studies and caution should be paid in the use of antioxidant supplements in the context of FGF/FGFR inhibitory therapies in lung cancer.

## 4. Materials and Methods

### 4.1. Cell Cultures and Reagents

Human H1581, H520 and HCC827 cell lines were obtained from ATCC and cultured in RPMI plus 10% FBS. Murine TC-1 cells were kindly provided by L. Accardi (Istituto Superiore Sanità, Rome, Italy) and cultured as described [34]. Cells were maintained at low passage, returning to original frozen stocks every 3 to 4 months, were tested regularly for mycoplasma negativity and were characterized for the expression of the different members of FGF/FGFR system (Appendix A). For lentiviral transduction of H1581 and H520 cells, particles coding for the mitochondrial catalase (pLVX-EF1a-IRES-puro-CatalaseMito) or the cytoplasmic catalase (pLVX-IRES-neo-hCatalaseCito) were used [35]. Erdafitinib was obtained from Selleckchem (TX, USA) and NSC12 was synthesized as previously described [10]. Vitamin E was obtained from Merck (Milan, Italy).

### 4.2. Cytofluorimetric Analyses

Cytofluorimetric analyses were performed using the MACSQuant^®^ Analyzer (Miltenyi Biotec, Bergisch Gladbach, Germany). Propidium iodide staining (Immunostep, Salamanca, Spain) was used to detect PI negative viable cells and viable cell counts were obtained by the counting function of the MACSQuant^®^ Analyzer. Mitochondrial membrane depolarization, mitochondrial (mt)ROS production and cytoplasmic ROS levels were determined using the fluorescent probes tetramethylrhodamine ethyl ester (TMRE), Mitosox and DCF-DA (Thermo Fischer Scientific, Monza, Italy), respectively. Apoptotic cell death was assessed by Annexin-V/propidium iodide double staining (Immunostep, Spain) according to manufacturer’s instructions. For all the cytofluorimetric assays (apoptosis, TMRE, Mitosox and DCF-DA), the whole cell population (viable and dead cells) was analyzed.

### 4.3. Western Blot Analysis

Cells were washed in cold PBS and homogenized in NP-40 lysis buffer (1% NP-40, 20 mM Tris–HCl pH 8, 137 mM NaCl, 10% glycerol, 2 mM EDTA, 1 mM sodium orthovanadate, 10 μg/mL aprotinin, 10 μg/mL leupeptin). Protein concentration in the supernatants was determined using the Bradford protein assay (Bio-Rad Laboratories, Milan, Italy). Blotting analysis was performed using anti-phospho-FGFR, anti-c-Myc and anti-phospho-H2AX (Cell Signaling Technology, MA, USA). To normalize the amount of loaded proteins, all blots were probed with anti-α-tubulin (Merck, Milan, Italy) or anti-GAPDH (Santa Cruz Biotechnology, CA, USA) antibodies. Chemiluminescent signal was acquired by ChemiDoc™ Imaging System (Bio-Rad Laboratories, Milan, Italy).

### 4.4. RT-qPCR

Total RNA was extracted using TRIzol Reagent (Invitrogen, CA, USA) according to manufacturer’s instructions. Two μg of total RNA were retro-transcribed with MMLV reverse transcriptase (Invitrogen) using random hexaprimers. Then, cDNA was analyzed by quantitative PCR using primers for the FGF/FGFR family members as indicated in [22] and for the following c-Myc targets: *PRDX3For: ACAGCCGTTGTCAATGGAGAG, Rev: ACGTCGTGAAATTCGTTAGCTT; PRDX4 For: GCAAAGCGAAGATTTCCAAGC, Rev: CGCCAAAAGCGATAATTTCAGTT; EIF3B For: CTTACGGGGCACAAGAAGAG, Rev: CCACCAGGAGACCAAGAAAA; MCM2 For: AGGAGAGTCCAGGCAAAGTG, Rev: CAGTGGCAAAGACAGGGAAG. Human 18S* (*For: GGGACTTAATCAACGCAAGC, Rev: GCAATTATTCCCCATGAACG*) was used as housekeeping gene.

### 4.5. Subcutaneous Human Xenografts

Experiments were performed according to Italian laws (D.L. 116/92 and following additions) that enforce the EU 86/109 Directive and were approved by the local animal ethics committee (OPBA, Organismo Preposto al Benessere degli Animali, Università degli Studi di Brescia, Italy). H520 cells (5 × 10^6^ cells/implant) and TC-1 cells (1 × 10^5^ cells/implant) were injected in 7-week-old Nu/Nu or C57BL/6 female mice, respectively. When tumors were palpable, mice were randomly assigned to receive treatment with NSC12 (7.5 mg/kg) or control/vehicle DMSO and treatments were performed as previously described [21]. Tumor volumes were measured with calipers and were calculated according to the formula V = (D × d2)/2, where D and d are the major and minor perpendicular tumor diameters, respectively. At the end of the experimental procedure, tumor nodules were excised and processed for histological analysis. As already reported [21], treatment with NSC12 was devoid of any relevant side effect in vivo in terms of body weight variation and alteration of hematological and biochemical parameters.

### 4.6. Histological Analyses

Formalin-fixed, paraffin-embedded tumor samples were sectioned at a thickness of 3 μm, dewaxed, hydrated and stained with hematoxylin and eosin (H&E) or processed for immunohistochemistry with rabbit anti-human c-Myc (Abcam), rabbit anti-pH2AX (Cell Signaling Technology, MA, USA) or rabbit anti-nitrotyrosine (Millipore) antibodies.

Positive signal was revealed by 3,3’-diaminibenzidine (Roche) stainings. Sections were finally counterstained with Carazzi’s hematoxylin before analysis by light microscopy. Images were acquired with the automatic high-resolution scanner Aperio System (Leica Biosystems, Wetzlar, Germany, EU) and image analysis was carried out using the open-source ImageJ software.

### 4.7. Statistical Analyses

Statistical analyses were performed using Prism 8 (GraphPad 6.0 Software, CA, USA). Student’s t test for unpaired data (2-tailed) was used to test the probability of significant differences between two groups of samples. For more than two groups of samples, data were analyzed with a 1-way analysis of variance and corrected by the Bonferroni multiple comparison test. Tumor volume data were analyzed with a 2-way analysis of variance and corrected by the Bonferroni test. Differences were considered significant when *p* < 0.05.

## Figures and Tables

**Figure 1 ijms-21-09376-f001:**
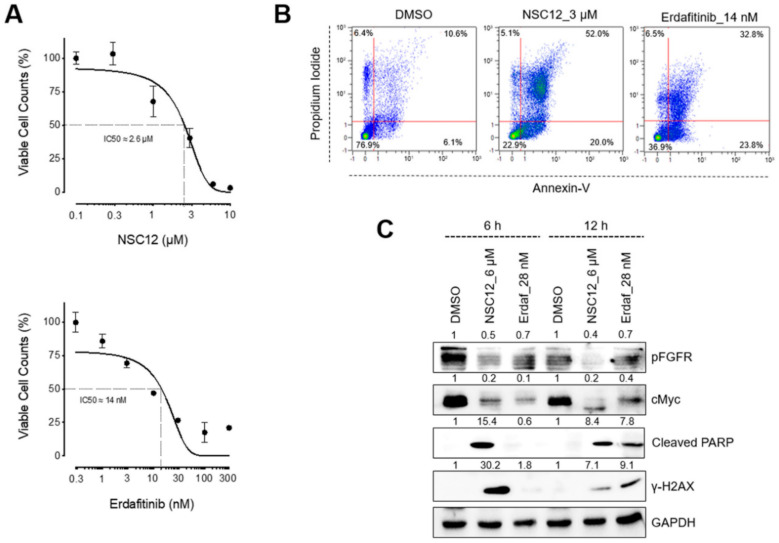
FGF (fibroblast growth factor)/FGFR (FGF receptor) inhibition induces apoptosis and DNA damage in FGF-dependent lung cancer cells. (**A**) H1581 viable cell counts by flow cytometric analysis 48 h after treatment with increasing doses of NSC12 or erdafitinib. Data are mean ± SEM of 3 or more experimental replicates. (**B**) Cytofluorimetric analysis of H1581 cell apoptosis by propidium iodide/Annexin-V double staining 48 h after treatment with NSC12 or erdafitinib. (**C**) Western blot analysis of H1581 cells after 6 and 12 h of treatment with NSC12 or erdafitinib. Densitometric quantification is reported above each band.

**Figure 2 ijms-21-09376-f002:**
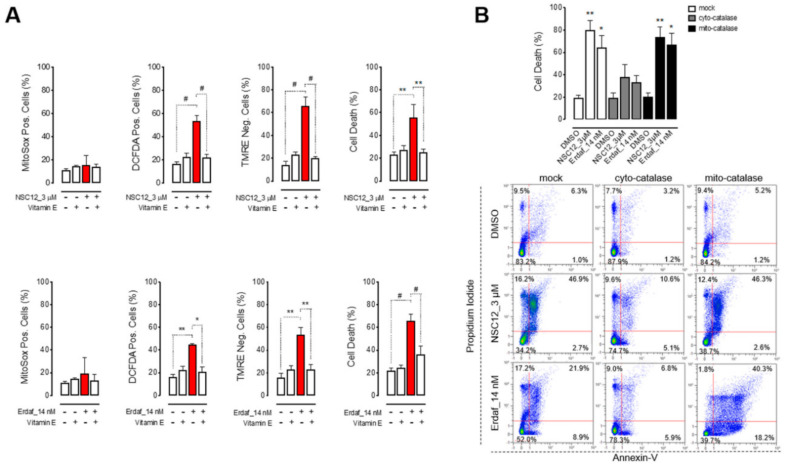
Apoptosis upon FGF/FGFR inhibition is mediated by oxidative stress. (**A**) H1581 cells were treated with NSC12 or erdafitinib in presence or absence of vitamin E (220 μM) for 48 h and cytofluorimetric analyses for mitochondrial or cytoplasmic ROS production, mitochondrial membrane depolarization and apoptosis by Mitosox, DCF-DA, TMRE and propidium iodide/Annexin-V stainings, respectively, were performed. (**B**) Upper panel: Percentage of mock and mitochondrial or cytoplasmic catalase-overexpressing H1581 cell death (calculated as the sum of Annexin-V+/PI-, Annexin-V+/PI+, Annexin-V-/PI+) after treatment with NSC12 or erdafitinib for 48 h. Lower panel: representative dot plots from cytofluorimetric analysis. Data are mean ± SEM of 3 or more experimental replicates. * *p* < 0.05, ** *p* < 0.01, # *p* < 0.001.

**Figure 3 ijms-21-09376-f003:**
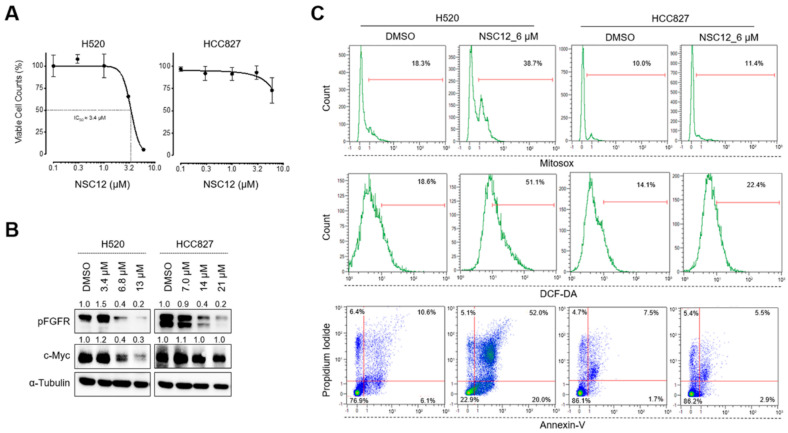
c-Myc downregulation and oxidative stress by FGF blockade is specific for FGF-dependent lung cancer cells. (**A**) H520 and HCC827 viable cell counts by flow cytometric analysis 48 h after treatment with increasing doses of NSC12. Data are mean ± SEM of 3 or more experimental replicates. (**B**) Western blot analysis of H520 and HCC827 cells after 3 h of treatment with increasing doses of NSC12. Densitometric quantification is reported above each band. (**C**) Cytofluorimetric analyses for mitochondrial or cytoplasmic ROS (reactive oxygen species) production and apoptosis by Mitosox, DCF-DA and propidium iodide/Annexin-V stainings, respectively.

**Figure 4 ijms-21-09376-f004:**
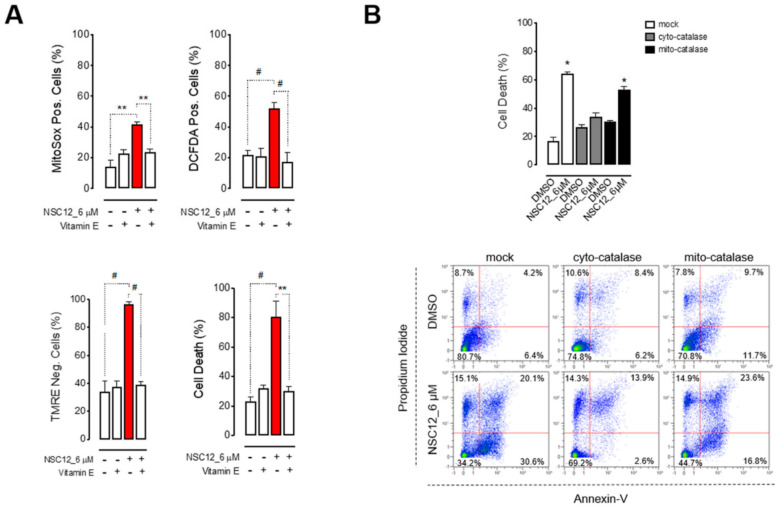
FGF trapping induces oxidative-stress-mediated apoptosis in FGF-dependent lung cancer cells. (**A**) H520 cells were treated with NSC12 in presence or absence of vitamin E (220 μM) for 48 h and cytofluorimetric analyses for mitochondrial or cytoplasmic ROS production, mitochondrial membrane depolarization and apoptosis by Mitosox, DCF-DA, TMRE and propidium iodide/Annexin-V stainings, respectively, were performed. (**B**) Upper panel: Percentage of mock and mitochondrial or cytoplasmic catalase-overexpressing H520 cell death (calculated as the sum of Annexin-V+/PI−, Annexin-V+/PI+, Annexin-V−/PI+) after treatment with NSC12 for 48 h. Lower panel: representative dot plots from citofluorimetric analysis. Data are mean ± SEM of 3 or more experimental replicates. * *p* < 0.05, ** *p* < 0.01, # *p* < 0.001.

**Figure 5 ijms-21-09376-f005:**
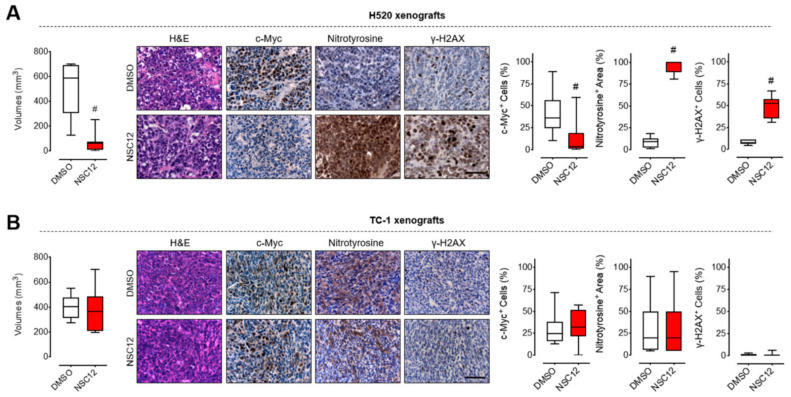
In vivo c-Myc modulation, oxidative stress and DNA damage mediated by FGF trapping are specific for FGF-dependent lung cancers. (**A**) FGF-dependent H520 tumor xenografts. (**B**) FGF-independent TC-1 tumor xenografts. Left panels: volumes of subcutaneous tumors 23 days post-implantation treated with NSC12 (7.5 mg/kg) or control/vehicle DMSO. Middle panels: histological analyses of DMSO- or NSC12-treated tumors 23 days post-implantation. Right panels: quantification of c-Myc, nitrotyrosine and γ-H2AX positivity by using ImageJ software. In box and whiskers graphs, boxes extend from the 25th to the 75th percentiles, lines indicate the median values and whiskers indicate the range of values. *n* = 10 mice/group, # *p* < 0.001.

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
