# Peer review of "Inhibition of the FGF/FGFR System Induces Apoptosis in Lung Cancer Cells via c-Myc Downregulation and Oxidative Stress"

_ijms, 2020, doi:10.3390/ijms21249376_

Round 1
Reviewer 1 Report
The manuscript by Giacomini et al. entitled “Inhibition of the FGF/FGFR system induced apoptosis in lung cancer cells via c-Myc down-regulation and oxidative stress” is an interesting and important study examining the effect of blocking the FGF/GFGR system by tyrosine kinase inhibitors or FGF-ligand inhibitors as a therapeutic approach to lung cancers. Overall the study is well-written and informative. The inclusion of the tumor xenograft data provides strong evidence for the authors conclusions. However, a few clarifications need to be address as outlined below.
- The decrease in c-Myc to critical for the authors conclusions and thus needs to be quantitated in figure 1 and figure 3.
- The authors need to clarify what is meant by “TMRE negative cells” as this dye as a measure of mitochondrial membrane potential would show shifts in the mean fluorescence intensity (MFI) as opposed to a clear positive/negative population of cells. Including a representative plot of this as well as for the MitoSox and DCFDA analysis would be quite beneficial.
- In regards to the comments above, it is unclear if the authors are analyzing only the viable population of cells. In the Annexin/PI plots shown in figures 1-4, there is a clear PI positive/Annexin negative (upper left-hand quadrant) that should be relatively absent in this assay; if the cells have lost their membrane integrity and thus PI positive, Annexin should be able to stain these cells by getting inside the cells also. This needs additional clarification.
- The number of experiments used for statistical analysis needs to be stated for all figures.
Reviewer 2 Report
The Authors presented an interesting article about the role of cMyc and oxidative stress in response to FGF/FGFR inhibition using TKIs and FGF trapping (NSC12) in FGFR-driven NSCLC. The article is overall well written and pleasant to read.
The article is highly relevant since several therapeutic approaches of FGF/FGFR blockade are under evaluation for the treatment for this subtype of lung cancer.
Nonetheless I have two major concerns:
- Novelty: Giacomini et al. published already similar finding in FGFR-driven multiple myeloma. (Ronca et al. FGF Trapping Inhibits Multiple Myeloma Growth through c-Myc Degradation-Induced Mitochondrial Oxidative Stress. Cancer Res 2020 Jun 1;80(11):2340-2354. doi: 10.1158/0008-5472.CAN-19-2714.)
- The article here presented is rather descriptive and do not provides a deeper mechanistic insight.
General comments:
- The number of biological replicates of each experiment, as well as the number of mice/group in the in vivo experiments should be indicated in the legend.
Specific comments:
Fig1.
A blot for cleaved Caspase3 or cleaved PARP should be added.
pH2AX is phospho-Ser139? If this is the case, it is generally better known as γ-H2AX. If this is the case it would e better to modify it everywhere.
How cMyc target genes are modulated upon NSC12 treatment? qPCR analyses of some cMyc target genes could be informative.
Fig.3
The amount of secreted FGF should be measured in each cell line (FGF-dependent and independent). Either total FGF or the most abundant isoforms. Alternatively, FGF mRNA level should be provided.
Figure4.
Why TC-1 xenograft in Fig4B? This cell line does not appear elsewhere in the paper. To keep consistency all along the paper please consider using HCC827 instead. Alternatively please provide in vitro data (Mitosox, DCFDA, Apoptosis and Viability) for TC-1cells as supplemental data.
Does NSC12 binds human and murine FGF with the same affinity? This point should be mentioned in the text.
Could it be possible to plot the whole kinetics of tumor growth/inhibition instead of box and whiskers graphs (A and B left panels).
How many mice per group? Should be indicated in figure legend.
Does NSC12 treatment cause any side effects? Mouse weight under vehicle or NSC12 treatment should be presented to prove treatment safety. Do authors observer any side effect in mice upon FGF-trapping?
Does NSC12 treatment affect the tumor microenvironment? If authors dispose of xenograft tumors paraffin-embedded blocs an analysis of the tumor microenvironment might reveal interesting differences and add some degree of novelty to the paper. For example, IHC analysis of CD45, CD31, α-SMA should be performed.
Round 2
Reviewer 2 Report
The authors answered all comments. Therefore I would recommend the new revised version of the manuscript for publication.